# Sustaining the Continued Effectiveness of an Antimicrobial Stewardship Program in Preterm Infants

**DOI:** 10.3390/tropicalmed9030059

**Published:** 2024-03-07

**Authors:** Tommaso Zini, Francesca Miselli, Chiara D’Esposito, Lucia Fidanza, Riccardo Cuoghi Costantini, Lucia Corso, Sofia Mazzotti, Cecilia Rossi, Eugenio Spaggiari, Katia Rossi, Licia Lugli, Luca Bedetti, Alberto Berardi

**Affiliations:** 1Neonatal Intensive Care Unit, Department of Medical and Surgical Sciences of Mothers, Children and Adults, University of Modena and Reggio Emilia, 41125 Modena, Italyalberto.berardi@unimore.it (A.B.); 2PhD Program in Clinical and Experimental Medicine, University of Modena and Reggio Emilia, 41125 Modena, Italy; 3Degree Program in Medicine and Surgery, University of Modena and Reggio Emilia, 41125 Modena, Italy; 4Post-Graduate School of Paediatrics, Department of Medical and Surgical Sciences of Mothers, Children and Adults, University of Modena and Reggio Emilia, 41125 Modena, Italy

**Keywords:** early onset sepsis, late-onset sepsis, antimicrobial stewardship, antibiotic use, clinical audit, newborn, very low birth weight, extremely low birth weight, neonatal intensive care unit

## Abstract

Background: There are wide variations in antibiotic use in neonatal intensive care units (NICUs). Limited data are available on antimicrobial stewardship (AS) programs and long-term maintenance of AS interventions in preterm very-low-birth-weight (VLBW) infants. Methods: We extended a single-centre observational study carried out in an Italian NICU. Three periods were compared: I. “baseline” (2011–2012), II. “intervention” (2016–2017), and III. “maintenance” (2020–2021). Intensive training of medical and nursing staff on AS occurred between periods I and II. AS protocols and algorithms were maintained and implemented between periods II and III. Results: There were 111, 119, and 100 VLBW infants in periods I, II, and III, respectively. In the “intervention period”, there was a reduction in antibiotic use, reported as days of antibiotic therapy per 1000 patient days (215 vs. 302, *p* < 0.01). In the “maintenance period”, the number of culture-proven sepsis increased. Nevertheless, antibiotic exposure of uninfected VLBW infants was lower, while no sepsis-related deaths occurred. Our restriction was mostly directed at shortening antibiotic regimens with a policy of 48 h rule-out sepsis (median days of early empiric antibiotics: 6 vs. 3 vs. 2 in periods I, II, and III, respectively, *p* < 0.001). Moreover, antibiotics administered for so-called culture-negative sepsis were reduced (22% vs. 11% vs. 6%, *p* = 0.002), especially in infants with a birth weight between 1000 and 1499 g. Conclusions: AS is feasible in preterm VLBW infants, and antibiotic use can be safely reduced. AS interventions, namely, the shortening of antibiotic courses in uninfected infants, can be sustained over time with periodic clinical audits and daily discussion of antimicrobial therapies among staff members.

## 1. Introduction

Sepsis threatens the survival of neonates and is associated with unfavourable long-term outcomes [1,2]. Strategies to prevent neonatal sepsis, its early diagnosis, and timely treatment are essential to reduce its incidence, mortality, and sepsis-related complications [3,4]. However, clinical presentation may be nonspecific, reliable early markers of sepsis are currently unavailable, and risk factors have insufficient predictive value [5,6]. In preterm very-low-birth-weight (VLBW) infants, both the incidence of sepsis and the associated risks of mortality or complications are higher than in full-term infants [1,2]. Antibiotics play a key role in the timely treatment of early and late-onset sepsis (EOS and LOS, respectively) with both empiric and targeted antibiotic therapy [1,2,7]. It is therefore not surprising that antimicrobials are the drugs most commonly prescribed by neonatologists and that most VLBW infants receive prolonged courses of antibiotics during their hospital stay in the neonatal intensive care unit (NICU) [8]. Approximately 14% of late preterm and full-term neonates and up to 90% of extremely-low-birth-weight (ELBW) infants receive empirical antibiotics immediately after birth, although culture-proven EOS occurs in only a few of these neonates [8]. Antibiotic exposure during the first few days of life may have important adverse effects [9,10]. Indeed, early and prolonged exposure to antibiotics may increase the risk of antimicrobial resistance and lead to dysbiosis, with a potential negative impact, especially in preterm infants [8,9,10]. Short- and long-term morbidity after antibiotic exposure has been reported in uninfected preterm infants [8,9,10]. For example, early and prolonged administration of antibiotics has been associated with an increased risk of late-onset sepsis (LOS), necrotizing enterocolitis (NEC), bronchopulmonary dysplasia (BPD), intraventricular haemorrhage (IVH), retinopathy of prematurity (ROP), invasive candidiasis, and death [9,10]. Early antibiotic use is associated with greater late antibiotic use [10]. Moreover, abnormal development of the intestinal flora during the first weeks of life may increase the risk of adverse long-term outcomes, including diabetes, obesity, asthma, inflammatory bowel disease (IBD), neurodevelopmental disorders, atopic dermatitis, and multiple sclerosis [11]. Prolonged and repeated antibiotic courses in uninfected infants accelerate the development and spread of antimicrobial resistance (AMR) and increase the risk of morbidity and mortality due to multidrug-resistant organisms (MDROs), as empiric therapy is less likely to be effective against these pathogens [9,10,12]. Infections caused by MDROs are also associated with prolonged hospitalization, increased costs, and death [12].

Wide variations exist in empirical antibiotic use in NICUs despite comparable infection rates and antibiotic resistance profiles [13,14,15,16]. There are many reasons for this, including limited pharmacokinetic and pharmacodynamic data in preterm and term infants and few high-quality clinical trials on which expert consensus or international guidelines for treatment and prophylaxis can rely [14,15]. However, comparing one’s NICU with population-based studies from different regions and centres may lead to revisiting one’s work and reconsidering one’s approaches [15,16].

Antimicrobial stewardship (AS) refers to a strategy or set of interventions to improve and measure the responsible and appropriate use of antimicrobial agents through the promotion of optimal drug selection, dosage, duration, and route of administration, as well as control of antibiotic use [17,18]. AS programs aim to achieve the best clinical outcomes for the treatment or prevention of infection, with minimal unintended consequences of antimicrobial use, including adverse events and toxicity to the patients and the emergence of resistance [17,18]. Ongoing surveillance of antibiotic consumption is crucial to any AS program [8,17]. In addition, a correlation between the burden of treatment and the burden of disease contributes to optimizing effective sepsis management and antimicrobial stewardship [8,17]. Several methods are used to quantitatively measure the extent of antibiotic use in clinical practice, including the days of antibiotic therapy (DOT) per 1000 patient days (PD, DOT/1000 PD) and the antibiotic use rate (AUR) per 1000 patient days (AUR/1000 PD) [19]. The first (DOT/1000 PD) is one of the most widely used measures and sums the duration of exposure to each antimicrobial drug; indirectly, it may incentivize the use of broad-spectrum monotherapies over narrow-spectrum combination therapies [19]. The second (AUR/1000 PD) depends on the days of exposure to any antibiotic relative to total inpatient days, regardless of the choice of drug(s) [19].

Studies on AS in preterm and term infants are limited. VLBW infants are those at highest risk of sepsis and who benefit most from judicious use of antibiotics, especially in the first week of life, but the impact and safety of AS programs on preterm neonates are not clearly defined [3,14,18]. Quality improvement initiatives in NICUs have been proven to be effective in the short term [20,21]. In particular, there is short-term evidence that antibiotic use immediately after birth can be reduced with systematic implementation of an AS program aimed at avoiding antibiotic exposure in preterm infants without perinatal EOS risk factors [22]. In the short term, the feasibility of AS programs has been demonstrated both in industrialized and developing countries [23,24]. However, there are limited data on the long-term effects of systematic application of a standardized AS program, especially for infants of lower gestational age and lower birth weight [14].

We previously demonstrated that AS is feasible in our setting and that antibiotic use can be safely reduced in neonatal care with the systematic application of an AS program [3]. We have already described in detail the AS program in our NICU and the benefits reached in VLBW infants after 2 years of multidisciplinary meetings and training courses [3]. The purpose of the present study is to assess if changes in antibiotic use have occurred over time and whether the effects of our AS program were maintained by more targeted antibiotic therapies and continuous surveillance of neonatal outcomes. Secondly, we aim to assess whether changes in protocol and algorithms and subsequent reduced antibiotic use led to any clinical worsening, namely, sepsis-related mortality.

## 2. Materials and Methods

### 2.1. Setting, Data Source, and Ethics

In the province of Modena, Italy, the resident population remained stable (from 2011 to 2021) at approximately 701,000 people, according to the report by Istat (the National Statistics Institute). Public health care is free for all residents. Currently, in Italy, almost all births take place in public hospitals. Hospital services in the province of Modena are organised into three health care areas (Central, North, and South), with three respective main birthing centres, in one of which there is the referral NICU at the University Hospital of Modena. VLBW infants are admitted to our level III NICU. Data for this study were collected retrospectively by accessing the Vermont Oxford Network (VON) database (in which our centre’s data are entered, according to privacy regulations), NICU computerised medical records (Metavision Suite, iMDSOFT, version 5.40.44, Tel Aviv, Israel), and nursing flowsheets. Data were obtained by surveillance officers using a standardized form. The Local Ethics Committee Area Vasta Emilia Nord (AVEN) approved this study (Protocol AOU 0002163/19 as amended).

### 2.2. Study Design

This observational, retrospective, single-centre study was carried out at the NICU of the University Hospital of Modena, Italy. The characteristics of our NICU have not changed substantially from what was described in a previous article [3]. The study population consists of VLBW infants who were admitted to our NICU during three 2-year periods: I. the “baseline period”, before the antimicrobial stewardship (AS) program, from 1 January 2011 to 31 December 2012 (live births *n* = 6744, VLBW infants *n* = 111, 1.6%); II. the “intervention period”, after implementation of the AS program, from 1 January 2016 to 31 December 2017 (live births *n* = 5902, VLBW infants *n* = 119, 2.0%); and III. the “maintenance period”, approximately 5 years later, from 1 January 2020 to 31 December 2021 (live births *n* = 5861, VLBW infants *n* = 100, 1.7%). In this study, we expanded the results of the previous article by extending the analyses to a third period so that we could evaluate the maintenance of the effects of our AS program over about 5 years. The AS implementation interventions between “baseline” and “intervention” have already been described in the previous article [3]. The same protocols and algorithms to guide the use of antibiotics developed earlier have also been implemented between the “intervention” and “maintenance” periods. Periodic clinical audits have been continued, but dedicated training courses for medical and nursing staff have been reduced, and the COVID-19 pandemic has further undermined continuing medical education on topics other than COVID-19 during 2020–2021.

Some key elements of our AS program were already in place in the “intervention period” and were implemented in the “maintenance period”. We increased the diagnostic sensitivity of blood cultures via the routine collection of double blood cultures (for both aerobes and anaerobes) [1,2]. In addition, the poor predictive value of abnormal biomarkers was discussed in clinical audits, and the routine use of C-Reactive Protein was reduced, particularly as a screening test for suspected EOS [25,26]. As a result, we reduced cases in which the initiation and continuation of antibiotics were strictly based on laboratory criteria. Moreover, the AS team was increasingly involved in decision making regarding antibiotic therapies [27]. Our strategies to shorten the antibiotic courses in VLBW infants with negative cultures were also further implemented. Urine cultures were almost always obtained via sterile catheters, whereas routine respiratory and gastrointestinal viral testing (CMV, others) were routinely performed [27].

Together with measures to reduce the use of unnecessary antibiotics, in the “maintenance period”, we ensured greater and immediate protection of infants at greatest risk of sepsis-related mortality and morbidity by improving empirical treatment regimens for severe sepsis and septic shock [28]. If meningitis could not be ruled out through a diagnostic lumbar puncture (because of haemodynamic instability), empiric antimicrobials with high delivery through the blood–brain barrier (including add-on third-generation cephalosporins) were administered at a very early stage, pending culture results [28].

In the clinical practice of our centre, further protocols have been updated, and some quality improvement initiatives on different topics may have had an impact on AS. The treatment of neonatal shock, including septic shock, has been revised, with an emphasis on early hemodynamic support. Similarly, we updated the management of central venous catheters (CVCs) and the preparation of parenteral nutrition, which has a potential impact on sepsis and antibiotic use. In addition, the “maintenance period” was during the COVID-19 pandemic. As for VLBW infants in our NICU, the SARS-CoV-2 outbreak mainly led to increased adherence to existing preventive measures. However, we adapted some protocols, including (i) hygiene measures, (ii) parents’ access to the NICU, and (iii) periodic viral screening. We identified the following as the major changes: (i) increased use of face masks, (ii) stricter rules for visitors (although parents’ access was only transiently limited), and (iii) respiratory viral testing for all symptomatic infants. Therefore, COVID-19 prevention regulations may be considered to have had an impact as further AS interventions in the “maintenance period”.

### 2.3. Exclusion Criteria

The use of topical antibiotics was excluded from the analysis. Moreover, VLBW infants who were referred to our NICU but had an uncertain history of parenteral antibiotic use (e.g., out-born infants with short hospitalization at our centre or inborn infants early transferred to another centre) were excluded.

### 2.4. Data Collection

The following demographic characteristics were evaluated: gender, gestational age, birth weight, singleton or twin pregnancies, 5 min Apgar score, the CRIB (clinical risk index for babies) score, maternal indication for delivery, and mode of delivery. The risk factors for EOS considered included chorioamnionitis, prolonged rupture of membranes (PROM > 18 h), maternal fever in labour (T > 38 °C), positive maternal Group B Streptococcus (GBS) screening, and intrapartum antibiotic prophylaxis. We considered the length of hospital stay and the overall duration of CVC placement. Data were extracted on the use of all antibiotics, excluding those administered topically. Narrow-spectrum β-lactams included benzylpenicillin, ampicillin and oxacillin, while aminoglycosides included gentamicin and amikacin. These are the two groups of antibiotics most used in combination therapy in our NICU. In another group defined as broad-spectrum β-lactams, we included third-generation cephalosporins and carbapenems, which we distinguished from generation I and II cephalosporins. The glycopeptide group included teicoplanin and vancomycin. The following data related to antibiotic use were recorded: the timing of the initiation of the first antibiotic treatment and its duration, the drug used as the first course and any reinstitution and, finally, the overall duration of antibiotic therapies during hospitalization. Antibiotic use was measured quantitatively as both DOT/1000 PD and AUR/1000 PD.

Since antibiotic consumption also depends on infection rates, we analysed the cases of sepsis in the study population, including EOS, LOS, sepsis with coagulase-negative staphylococci (CoNS), and culture-negative sepsis. We also recorded the cases of necrotizing enterocolitis (NEC) and death (sepsis-related mortality and in-hospital mortality).

### 2.5. Definitions

Extremely-low-birth-weight (ELBW) and very-low-birth-weight (VLBW) infants: neonates with birth weight (BW) less than 1000 g and 1500 g, respectively.

Culture-proven early onset sepsis (EOS) and late-onset sepsis (LOS): clinical signs of sepsis and isolation of a pathogen from blood or cerebrospinal fluid within the first 3 days of life (≤72 h of life, EOS) and after 3 days of life (>72 h of life, LOS), respectively, and antibiotic treatment for at least 5 days, or death within 5 days of treatment.

Culture-negative sepsis: sterile cultures, abnormal biomarker values, and clinical signs of sepsis in infants treated for ≥5 days with antibiotics but with sterile cultures.

Sepsis with coagulase-negative staphylococci (CoNS): clinical signs of sepsis, isolation of a CoNS from two blood cultures collected within 48 h (to distinguish from contamination), or administration of active antibiotics (e.g., glycopeptide or semisynthetic penicillin) for ≥5 days, according to the physician’s judgement.

Sepsis-related death: death occurring within 7 days of a positive blood culture or clearly related to the complications of sepsis.

Forty-eight hour rule-out sepsis: administration of empiric antibiotics ≤48 h pending culture results.

### 2.6. Statistical Analyses

Statistical analyses were performed using STATA/SE 14.2 (StataCorp, Lakeway, TX, USA). For descriptive data, non-parametric continuous variables were summarized as medians with interquartile ranges (IQRs, 25th and 75th percentiles), and categorical variables were presented as proportions (%, rates). Differences between groups were analysed with non-parametric (Kruskal–Wallis or Mann–Whitney U-test) or parametric (ANOVA and *t*-test) tests for continuous variables and Fisher’s exact or chi-squared tests for categorical data, as appropriate. Two-tailed *p* values of <0.05 were considered statistically significant.

## 3. Results

A total of 330 VLBW infants were enrolled, including 111, 119 and 100 in “baseline”, “intervention”, and “maintenance” periods, respectively (Figure 1). A few VLBW infants were excluded for missing data on antibiotic treatments: 8 infants in the “baseline period” (out-born infants with short hospitalization at our centre, *n* = 2; inborn infants transferred early to another centre, *n* = 6), 2 infants in the “intervention period” (out-born infants with short hospitalization at our centre, *n* = 1; inborn infants early transferred to another centre, *n* = 1), no infants in the “maintenance period”.

### 3.1. Demographics of VLBW Infants

Table 1 shows the prenatal, intrapartum, and postpartum data of VLBW infants (median gestational age 29 weeks, median birth weight 1.1 kg) in “baseline”, “intervention”, and “maintenance” periods. Baseline characteristics were similar in the three periods. Cases of exposure to intrapartum antibiotic prophylaxis (IAP) for the prevention of early onset GBS disease increased progressively in the “intervention” and “maintenance” periods. It is likely that there was an increase in the number of clinical diagnoses of “Triple I” by obstetricians, although the maternal GBS screening positivity rate remained unchanged (*p* = 0.76), and there was no increase in histologically diagnosed cases of chorioamnionitis (*p* = 0.99).

### 3.2. Overall Antibiotic Use in VLBW Infants and the Policy of 48-Hour Rule-Out Sepsis Antibiotic Course

Approximately 75% of VLBW infants received at least one course of antibiotic therapy during their hospital stay (77%, 71%, and 80% in “baseline”, “intervention” and “maintenance” periods, respectively) (Table 1). In the “intervention period”, there was a 29% reduction in both DOT/1000 PD and AUR/1000 PD, but this reduction in overall antimicrobial drug consumption was not maintained over time. There was no significant downward trend in the proportion of neonates who received early empirical antibiotics immediately after birth (66%, 57% and 62% in “baseline”, “intervention”, and “maintenance” periods, respectively). However, early antibiotics were given for shorter periods, and the antibiotic burden in the first week of life was significantly reduced through the policy of 48 h rule-out sepsis (Figure 2). Both the duration of empirical antibiotics immediately after birth and the average days of antibiotic administration in the first week of life reduced in the “intervention period” and further reduced in the “maintenance period”. Median days of early empirical antibiotic treatment reduced from 6 to 3 and 2 in “baseline”, “intervention”, and “maintenance” periods, respectively (Table 1). Despite this reduction in early antimicrobial exposure, antibiotic resumption did not increase in the following two weeks.

### 3.3. Neonatal Outcomes and Infections

Table 1 also shows data on neonatal outcomes. After the AS program, both sepsis-related and overall mortality remained unchanged in the “intervention period”; of note, no sepsis-associated deaths were recorded in the “maintenance period”. Furthermore, no deaths were due to culture-proven sepsis that occurred within 14 days of discontinuation of antibiotic treatment. The number of EOS remained stable, while the overall number of LOS increased slightly in the “intervention” and “maintenance” periods. Furthermore, there was a significant increase in sepsis with CoNS in the “maintenance period” compared with both previous periods (7%, 2% and 14% in “baseline”, “intervention”, and “maintenance” periods, respectively, *p* < 0.001). The number and percentage of VLBW infants treated for culture-negative sepsis decreased significantly over time (22%, 11%, and 6% in “baseline”, “intervention”, and “maintenance” periods, respectively, *p* = 0.002).

### 3.4. Antibiotic Use According to Birth Weight

Among ELBW infants (median gestational age of 26 weeks, IQR of 25–27 weeks), antimicrobial drug consumption was higher than among infants with BW between 1000 and 1499 g (median GA 30 weeks, IQR 29–30), without relevant differences in baseline characteristics and risk factors for EOS (Table 2). Approximately 92% of ELBW infants were exposed to antibiotics during their hospital stay. Prescriptive appropriateness in the first week of life improved significantly by implementing the policy of a 48 h empiric antibiotic course immediately after birth (pending culture results), even in the smallest and youngest infants. The rates of ELBW exposed to antimicrobials immediately after birth were 82%, 76% and 60% in “baseline”, “intervention” and “maintenance” periods, respectively. The reduction in overall antibiotic exposure was most evident among infants with a BW of 1000–1499 g.

### 3.5. Antibiotics Exposure of VLBW Infants without Culture-Proven Sepsis

Table 3 shows antimicrobial exposure for reasons other than culture-proven sepsis. Antibiotics were administered because of rule-out sepsis courses and presumed culture-negative sepsis significantly reduced (both DOT/1000 PD and AUR/1000 PD). This reduction affected most infants with BW between 1000 and 1499 g. Days on antibiotics were significantly reduced in the first week of life for both infants with a BW < 1000 g and between 1000 and 1499 g.

### 3.6. Antimicrobial Drugs Administered

Figure 3 shows DOT/1000 PD for each antimicrobial drug administered in “baseline”, “intervention”, and “maintenance” periods. The overall use of narrow-spectrum β-lactams and aminoglycosides did not change over time. There was a decline in the ampicillin–gentamicin combination and a parallel increase in the oxacillin–amikacin combination treatment. The use of broad-spectrum β-lactams did not change. There was a downward trend in the use of piperacillin/tazobactam. In the glycopeptide group, teicoplanin use decreased dramatically from “baseline” to “intervention” periods, but vancomycin use increased from “intervention” to “maintenance” periods. The use of first- and second-generation cephalosporins was unchanged.

## 4. Discussion

AS programs provide guidance for the judicious use of antibiotics but also recommend an ongoing review of their use through the dissemination of results to staff and the scientific community. Staff are required to be continuously informed regarding the importance of proper antibiotic use in clinical practice. Short-term analyses are essential to assess the benefits of interventions, whereas long-term analyses allow for the survey of neonatal outcomes and implementation of AS programs in second- and third-level centres [29,30]. There are limited data on the long-term maintenance of AS interventions in VLBW infants.

We analysed the long-term effects of an AS program that had been previously implemented [3]. We achieved a reduction in antibiotic use mainly through early discontinuation of empirical antibiotic courses. Despite the challenges of regular education and staff training during the COVID-19 pandemic, we have shown that key best practices on antibiotic use can be safely maintained in the long term without increasing mortality or antibiotic reinitiation after an early discontinuation.

In detail, we demonstrated the benefits of implementing a policy of 48 h rule-out sepsis on both early and overall exposure to antibiotics. The safety of discontinuing antibiotics within 48 h when cultures are sterile has been previously demonstrated [31]. In some NICUs, an antibiotic automatic stop order (ASO) resulted in reduced antibiotic exposure [32,33]. We confirmed that our AS program and the 48 h rule-out sepsis policy (mainly combining narrow-spectrum β-lactams and aminoglycosides) can be implemented safely in the long term. Notably, unnecessary antibiotics were reduced in VLBW infants without culture-proven sepsis. Our goal was to reduce antibiotic exposure and the risk of antimicrobial resistance in vulnerable ELBW and VLBW infants while providing enhanced and immediate protection in cases of severe sepsis and septic shock. Although overall cases of sepsis did not decrease in the “intervention” and “maintenance” periods, the reduction in antibiotic exposure was achieved without an increased mortality. As a matter of fact, no sepsis-related deaths were recorded in the “maintenance period” when rule-out sepsis antibiotic courses were shorter. The sepsis-related zero mortality we achieved was also due to improved treatment of cases of severe sepsis and septic shock, with prompt intensive support from the onset. In addition, the need to resume antibiotic therapies within 14 days of discontinuation did not increase. We believe that these results demonstrate a substantial improvement in terms of prescriptive appropriateness. They were achieved mostly through periodic clinical audits during the study periods. In fact, although the COVID-19 pandemic reduced opportunities for continuing medical education, staff training on AS was still maintained through clinical audits.

Our AS program led to reduced antibiotic exposure for culture-negative sepsis and antibiotic courses administered immediately after birth. Several strategies were promoted to optimize antibiotic use in culture-negative sepsis in VLBW infants. Strategies included greater involvement of the AS team in daily decision making and frequent discussion of clinical cases among staff members. Neonatologists increasingly relied on sterile cultures and cases of culture-negative sepsis were reduced by our common strategy of discontinuing antibiotics when cultures are sterile (48 h after collection) [34].

Reducing antibiotic exposure in VLBW infants is a difficult goal to achieve. AS programs can be effective when tailored to preterm infants, focusing on reducing initiation and/or shortening the duration of antibiotic therapy [35]. We did not reduce the number of infants exposed to antibiotics immediately after birth, but rather the duration of antibiotic courses administered. In other words, our restriction was mainly related to the shortening of antibiotic regimens, including those administered immediately after birth, although we failed to reduce the number of premature infants with early exposure. Flannery et al. reported a high percentage of VLBW infants receiving early antibiotic initiation (78.6%) in 297 academic and community hospitals in the United States [36]. Over the entire 7-year study period (2009 to 2015), the investigators observed no differences in rates of infants exposed to antibiotics, although their rates (median of 62%) were higher compared to our centre. Similarly, we could not confirm a decrease of antibiotic exposure over time. There is no evidence that routine antibiotic use has a protective effect in preterm infants at low-risk of EOS [37]. It remains a challenge for future studies to target antibiotic therapy exclusively for the few infants with the highest risk [37].

A similar comparison can be made between our setting and that of multicentre studies with large sample sizes. A Chinese cohort study enrolled 21,540 infants admitted to 25 tertiary NICUs during 2015–2018 [10]. Eighty-five per cent (85.0%) of infants received antibiotics early [10]. For each additional day of early antibiotics, the authors reported an increased morbidity, particularly an increased risk of BPD [10].

The overall rate of VLBW infants exposed to antibiotics at any time during hospital stay is slightly lower in our centre (75% vs. 80%) than the one reported in a recent Norwegian nationwide study, although our overall antibiotic use during the hospital stay is higher (DOT/1000 PD 267 vs. 149) [14]. Among 5296 infants (with a comparable gestational age and birth weight), Norwegian investigators reported a reduction in DOT/1000 PD over 10 years [14]. Although the reasons for these discrepancies are unclear, repeated courses of antibiotics or prolonged therapies in cases of culture-positive sepsis would explain our results.

Regarding the choice of empirical antibiotic regimens, we maintained adherence to combination therapy with narrow-spectrum β-lactams and aminoglycosides [38]. Contrary to what we expected, the overall use of vancomycin increased from the “intervention” to the “maintenance” period. However, sepsis with CoNS increased in the “maintenance period”. Furthermore, an analysis of individual cases showed that this unintended result was also due to two outlier infants who received prolonged monotherapy for specific severe diseases.

Both DOT/1000 PD and AUR/1000 PD were reduced in infants with a BW between 1000 and 1499 g. This result was even more noticeable if data on antimicrobial drugs for the treatment of culture-proven sepsis were excluded. However, a parallel reduction in overall exposure to antibiotics was not achieved for ELBW infants, whose clinical signs are often nonspecific and sepsis-related incidence, mortality, and morbidity are higher [39].

The reason for the increasing rate of sepsis with CoNS is unclear. However, we reviewed individual cases. A reduced adherence to the central venous catheters (CVCs) protocols may have occurred, due to high turnover of nursing staff. Therefore, we are now implementing bundle checklists for CVC insertion and maintenance. Continuous updating on topics parallel to antimicrobial stewardship, including hemodynamic support for septic shock and management of central lines, can have confounding implications on the long-term analysis of an AS program’s effects.

This study has some important limitations. Firstly, this is a single-centre study in an Italian NICU. The results may not be generalizable to different settings (e.g., NICUs other than level III, developing countries, or countries with non-public health care). Second, our AS program and the policy of 48 h rule-out sepsis antibiotic course have been safely maintained and implemented over time, but the sample size is relatively small, so we need to be cautious about safety conclusions and benefits should be confirmed with larger, area-based studies. In addition, the small sample size and single-centre nature of this study are the main reasons why it is difficult to draw conclusions about possible confounding factors, such as, for example, the increased practice of IAP by obstetricians for the prevention of GBS infection. Third, the “maintenance period” (2020–2021) overlapped with the COVID-19 pandemic: we presented the possible implications of this, but confounding factors and additional changes between “baseline”, “intervention”, and “maintenance” periods may have occurred. Fourth, we analysed only short-term adverse neonatal outcomes and lacked data on the faecal microbiome–metabolome. Reducing unnecessary antibiotic use, particularly shortening antibiotic therapy regimens, can be improved through the continued implementation of diagnostic practices.

## 5. Conclusions

Antimicrobial stewardship is feasible in preterm VLBW infants. Its effects on the appropriate use of antimicrobial agents can be sustained over time through periodic clinical audits and retrospective short- and long-term analyses. Reducing unnecessary antibiotic use, mainly by shortening the duration of antibiotic courses, can be improved through the continued implementation of diagnostic practices, the review of individual cases, and discussion of therapies among staff members. Vigilance in ensuring neonatal outcomes is necessary.

## Figures and Tables

**Figure 1 tropicalmed-09-00059-f001:**
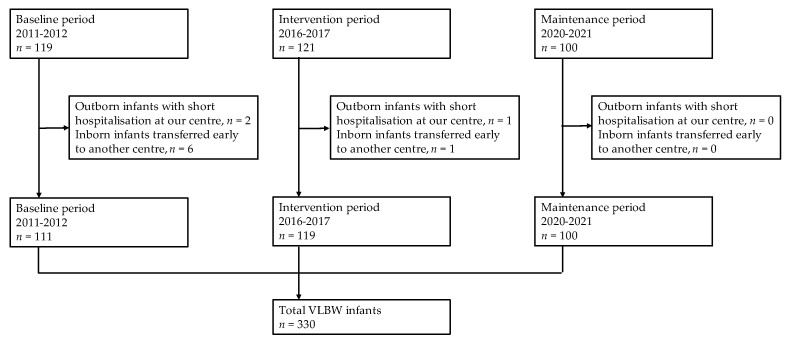
Study population.

**Figure 2 tropicalmed-09-00059-f002:**
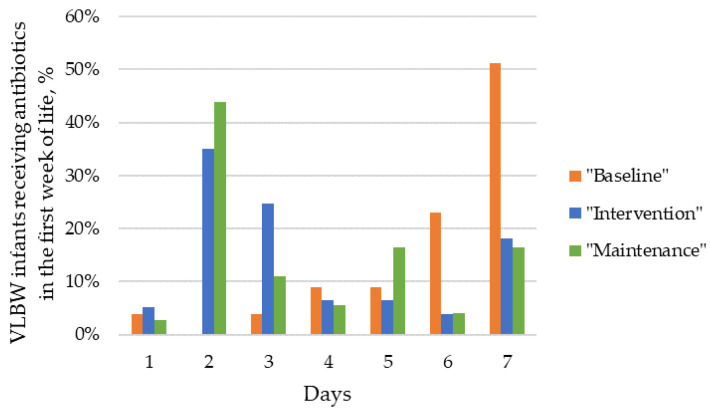
Early antibiotic exposure in VLBW infants in “baseline”, “intervention”, and “maintenance” periods: percentage of patients (*y*-axis) receiving *n* days of antibiotics (*x*-axis) in the first week of life.

**Figure 3 tropicalmed-09-00059-f003:**
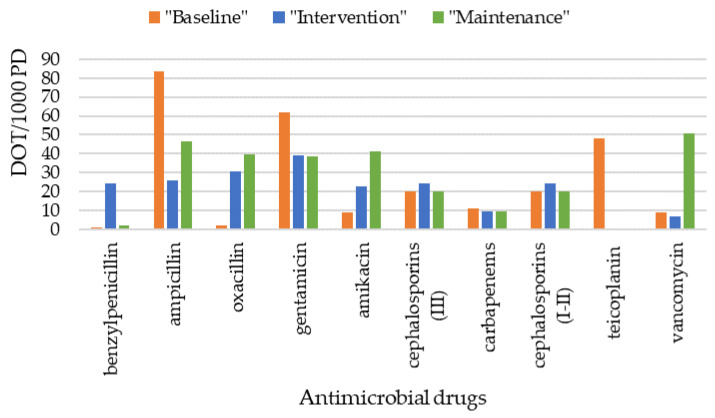
Antimicrobial drugs administered in “baseline”, “intervention”, and “maintenance” periods. Narrow-spectrum β-lactams include benzylpenicillin, ampicillin, and oxacillin. Aminoglycosides include gentamicin and amikacin. Broad-spectrum β-lactams include third-generation cephalosporins and carbapenems, which are distinguished from first- and second-generation cephalosporins. The glycopeptides group includes teicoplanin and vancomycin. Abbreviations—DOT/1000 PD: days of antibiotic therapy (DOT) per 1000 patient days (PD).

**Table 1 tropicalmed-09-00059-t001:** Demographic data and risk factors for early onset sepsis, antibiotics exposure, and neonatal outcomes in very-low-birth-weight infants in “baseline”, “intervention”, and “maintenance” periods.

	All VLBW Infants (*n* = 330)	“Baseline” 2011–2012 (*n* = 111)	“Intervention” 2016–2017 (*n* = 119)	“Maintenance” 2020–2021 (*n* = 100)	*p* Value
**Demographic data**					
Gender (male), *n* (%)	169 (51)	61 (55)	67 (56)	47 (47)	0.84
GA (weeks), median (IQR)	29 (26–31)	29 (26–31)	29 (26–31)	29 (27–31)	0.67
BW (g), median (IQR)	1148 (844–1370)	1146 (857–1346)	1109 (851–1398)	1188 (830–1339)	0.94
Twin birth, *n* (%)	75 (23)	28 (25)	31 (26)	16 (16)	0.16
5 min Apgar score, median (IQR)	8 (7–9)	8 (7–9)	8 (6–9)	8 (7–9)	0.27
CRIB score, median (IQR)	1 (0–4)	1 (1–4)	1 (0–4)	1 (1–4)	0.84
Maternal indication for delivery, *n* (%)	75 (23)	28 (25)	39 (33)	35 (35)	0.26
Mode of delivery, *n* (%) Vaginal delivery CS in labour or with membrane rupture CS before labour and with intact membranes	73 (22) 75 (23) 181 (55)	21 (19) 30 (27) 60 (54)	25 (21) 25 (21) 69 (58)	27 (27) 20 (20) 52 (52)	0.35 0.41 0.67
**Risk factors for EOS**					
Histological chorioamnionitis, *n* (%)	87 (26)	29 (27)	31 (26)	27 (27)	0.99
Prolonged rupture of membranes (PROM >18 h), *n* (%)	90 (27)	29 (26)	34 (29)	27 (27)	0.91
Maternal fever in labour (T > 38 °C), *n* (%)	14 (4)	3 (3)	6 (5)	5 (5)	0.61
Positive maternal GBS screening, *n* (%)	21 (6)	8 (7)	6 (5)	7 (7)	0.76
Intrapartum antibiotic prophylaxis, *n* (%) No Adequate	131 (40) 142 (43)	68 (63) 30 (28)	55 (48) 48 (40)	8 (8) 64 (64)	**<0.001** **<0.001**
**Exposures**					
Length of hospital stay (days), median (IQR)	47 (29–69)	47 (29–75)	46 (28–71)	48 (30–64)	0.97
Duration of CVC placement (days), median (IQR)	10 (4–25)	11 (4–27)	10 (4–28)	9 (4–20)	0.59
**Antibiotic use**					
DOT/1000 PD, *n*	267	302	215	291	0.09
AUR/1000 PD, *n*	162	192	136	160	0.07
No antibiotic exposure, *n* (%)	81 (25)	26 (23)	35 (29)	20 (20)	0.26
Empirical antibiotics immediately after birth, *n* (%)	203 (62)	73 (66)	68 (57)	62 (62)	0.40
Duration of early empirical antibiotic treatment (days), median (IQR)	5 (2–7)	6 (4–7)	3 (2–3)	2 (2–3)	**<0.001**
Days of antibiotics in the first week of life, median (IQR)	3 (0–7)	6 (0–7)	2 (0–5)	2 (0–5)	**<0.001**
**Outcomes**					
EOS, *n* (%)	9 (3)	3 (3)	4 (4)	2 (2)	0.83
LOS, *n* (%)	32 (10)	6 (5)	14 (12)	12 (12)	0.17
Sepsis with CoNS, *n* (%)	27 (8)	8 (7)	2 (2)	17 (17)	**<0.001**
Culture-negative sepsis, *n* (%)	43 (13)	24 (22)	13 (11)	6 (6)	**0.002**
Surgically treated NEC, *n* (%)	3 (1)	1 (1)	1 (1)	1 (1)	0.97
Sepsis-related mortality, *n* (%)	6 (2)	3 (3)	3 (3)	0 (0)	0.26
In-hospital mortality, *n* (%)	37 (11)	11 (10)	17 (14)	9 (9)	0.41

Abbreviations. AUR/1000 PD: antibiotic use rate (AUR) per 1000 patient days (PD); BW: birth weight; CoNS: coagulase-negative Staphylococci; CS: caesarean section; CVC: central venous catheter; DOT/1000 PD: days of antibiotic therapy (DOT) per 1000 patient days (PD); EOS: early onset sepsis; GA: gestational age; GBS: group B Streptococcus; IQR: interquartile range; LOS: late-onset sepsis; NEC: necrotizing enterocolitis.

**Table 2 tropicalmed-09-00059-t002:** Antibiotic exposure according to birth weight in “baseline”, “intervention”, and “maintenance” periods.

	BW < 1000 g (ELBW)				BW 1000–1499 g			
	“Baseline” 2011–2012 (*n* = 44)	“Intervention” 2016–2017 (*n* = 51)	“Maintenance” 2020–2021 (*n* = 35)	*p* Value	“Baseline” 2011–2012 (*n* = 67)	“Intervention” 2016–2017 (*n* = 68)	“Maintenance” 2020–2021 (*n* = 65)	*p* Value
**Risk factors for EOS**								
Histological chorioamnionitis, *n* (%)	20 (45)	20 (39)	12 (34)	0.48	9 (13)	11 (16)	15 (23)	0.32
Positive maternal GBS screening, *n* (%)	4 (6)	4 (6)	5 (8)	0.89	4 (9)	2 (4)	2 (6)	0.57
**Antibiotic use**								
DOT/1000 PD, *n*	367	269	449	0.62	237	154	182	**0.02**
No antibiotic exposure, *n* (%)	2 (5)	5 (10)	3 (9)	0.62	24 (36)	30 (44)	17 (26)	0.10
Empirical antibiotics immediately after birth, *n* (%)	36 (82)	39 (76)	21 (60)	0.08	37 (55)	29 (43)	41 (63)	0.06
Days of antibiotics in the first week of life, median (IQR)	7 (5–7)	3 (2–4)	2 (1–3)	**<0.001**	7 (5–7)	2 (2–3)	2 (1–3)	**<0.001**
**Outcomes**								
EOS, *n* (%)	3 (7)	3 (6)	2 (6)	0.97	3 (3)	4 (4)	2 (2)	0.83
LOS, *n* (%)	4 (9)	10 (20)	5 (14)	0.35	6 (5)	14 (12)	12 (12)	0.17
Sepsis with CoNS, *n* (%)	6 (14)	2 (4)	11 (31)	**0.002**	8 (7)	2 (2)	17 (17)	**<0.001**
Culture-negative sepsis, *n* (%)	15 (34)	9 (18)	4 (11)	**0.02**	24 (22)	13 (11)	6 (6)	**0.002**
Surgically treated NEC, *n* (%)	2 (5)	3 (6)	0 (0)	0.36	1 (1)	1 (1)	1 (1)	0.97
Sepsis-related mortality, *n* (%)	10 (23)	16 (31)	9 (26)	0.63	3 (3)	3 (3)	0 (0)	0.26
In-hospital mortality, *n* (%)	3 (7)	3 (6)	2 (6)	0.97	11 (10)	17 (14)	9 (9)	0.41

Abbreviations. BW: birth weight; CoNS: coagulase-negative Staphylococci; DOT/1000 PD: days of antibiotic therapy (DOT) per 1000 patient days (PD); ELBW: extremely low birth weight; EOS: early onset sepsis; IQR: interquartile range; LOS: late-onset sepsis.

**Table 3 tropicalmed-09-00059-t003:** Antibiotic exposure of VLBW infants without culture-proven sepsis in “baseline”, “intervention”, and “maintenance” periods.

	DOT/1000 PD				AUR/1000 PD			
	“Baseline” 2011–2012	“Intervention” 2016–2017	“Maintenance” 2020–2021	*p* Value	“Baseline” 2011–2012	“Intervention” 2016–2017	“Maintenance” 2020–2021	*p* Value
**VLBW infants without culture-proven sepsis**	251	134	166	**0.01**	158	85	91	**0.006**
ELBW infants without culture-proven sepsis	289	174	188	0.44	182	109	141	0.66
1000–1499 g BW infants without culture-proven sepsis	223	98	115	**0.004**	140	63	70	**0.002**

Abbreviations. AUR/1000 PD: antibiotic use rate (AUR) per 1000 patient days (PD); BW: birth weight; DOT/1000 PD: days of antibiotic therapy (DOT) per 1000 patient days (PD); ELBW: extremely low birth weight; VLBW: very low birth weight.

## Data Availability

De-identified individual participant data presented in this study are available on request from the corresponding author. The data are not publicly available due to the need for use in further research.

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
