# Peer review of "Sustaining the Continued Effectiveness of an Antimicrobial Stewardship Program in Preterm Infants"

_tropicalmed, 2024, doi:10.3390/tropicalmed9030059_

Round 1

Reviewer 1 Report

Comments and Suggestions for Authors

Nice and valuable manuscript. Maybe authors may propose widening of the investigation on a bigger number of NICUs. 

Reviewer 2 Report

Comments and Suggestions for Authors

I have read this paper with great interest.

I only have some minor, specific reflections to further improve the message of the paper.

It is not yet very clear in the abstract ‘where’ the reduction was: shorter courses, less courses, or less initiation at birth, EOS/LOS. Based on the full paper, it seems mainly related to the shortering of the ab use initiated at delivery. If so, this message should also be better reflected in the conclusions section of the paper.

As a second reflection, the statements on safety should perhaps be phrased somewhat more cautious, as the number in the 3 cohorts are overall rather small ?

Reviewer 3 Report

Comments and Suggestions for Authors

The authors describe the effects of a previously implemented antibiotic stewardship (AS) program after 4-5 years in a single-center retrospective cohort study, with focus on VLBW infants. They have previously published a publication describing the implemented AS actions and comparing baseline and intervention period. They achieved a reduction in antibiotic consumption mostly due early discontinuation of empiric antibiotic courses (ampicillin and gentamycin), initiated due to increased risk of early onset neonatal sepsis, while the profile of used antibiotics only slightly changed (teicoplanin was replaced with oxacillin). 

The current manuscript addresses antibiotic use in the “maintenance” phase, 4-5 years after the intervention of AS actions. Despite challenges of regular staff education and training due to the COVID-19 pandemic, both the duration of early empirical antibiotic treatment and days of antibiotics in the first week of life were comparable with results achieved in the implementation period; significantly lower compared with their baseline period. The reduction in total antibiotic consumption was most evident in infants with birth weight 1000 – 1499g, while antibiotic use was (as expected) higher in infants with birth weight < 1000g. The reduction in total antibiotic use achieved in intervention period was not maintained over time. No significant changes in the profile of antimicrobial drugs were achieved. 

Their study clearly shows the benefit of a 48-hour rule-out sepsis, a common AS action used for preterm infants, which was safely maintained over time, with no increase in mortality and no increase in resuming antibiotics 14 days after discontinuation. The effect was also observed in infants with birth weight <1000g. 

The abstract as it is now gives a false impression of the content of the article – since the focus is the maintenance period, this should be clear from the abstract (e.g. in the results section of the abstract, the numerical and statistical results are provided for intervention period only). I am also missing a mention of the 48-hour rule-out sepsis, as this is one of the strongest results of the study. 

The introduction should be shortened – especially lines 55-72 should be omitted, as they are a part of a broader topic not encompassed in the manuscript. The last two sections of the introductions are somewhat repetitive and should be clearly rewritten to state the aim of the current manuscript (lines 91-110). 

Baseline, intervention, and maintenance period are also referred to as I, II and III period – chose one phrasing and be consistent. 

In the method section, it is not clear which actions were implemented before and after intervention period. It appears that some new actions were implemented after the intervention period? E.g. double blood culture collection, reduction in routine use of CRP, improvement of empirical antibiotic therapy for severe sepsis, CVC management protocols, etc. A flow chart clearly stating actions implemented prior to intervention period and any additional actions implemented after intervention period (and prior to maintenance period) would give better clarity to the reader. If several actions were implemented AFTER the intervention period, this should me also discussed as a limitation in the discussion section.

The results are mostly reported as Days of therapy/1000 patient-days (DOT), which is a preferred metric in this setting compared with e.g. Defined daily dose/100 patient-days (DDD). The study also reports AUR/100 PD and duration of AB (days). Another metric that you might consider using is reporting the proportion of infants with negative blood cultures receiving prolonged antibiotic therapy (e.g. > 48 hours) between the three observational periods (similar as Ting et al. 2019). 

P values <0.05 should be bolded or marked with* in the tables. The X in Y axis should be clearly named in all figures. 

The graphics in Figures 2 and 3 should be improved for better clarity (the grey shades are very similar).

I do not understand the explanation regarding the increase in IAP in the maintenance period, this can perhaps be rephrased. Regarding histological diagnosis of chorioamnionitis – were all placentas sent for pathology, or only if chorioamnionitis was clinically suspected? Has there been any change in this practice (and in the practice of IAP prescription) during the study period? 

The first under-title (in my opinion) in the results section, after demographics, should be focusing on the effect of the 48-hour rule-out sepsis (and stratified according to BW), followed by the reduction in AB consumption in culture negative sepsis (and stratified according to BW), total AB consumption (and stratified by BW) and lastly the profiles of used antimicrobials. The sections in the discussion should follow the sections in the results section in the same order.

Are the infants in table 3 only infants with rule-out sepsis and infants with culture negative sepsis, or were there any other indication for antibiotic treatment of these infants? I think it would be more informative to state the antibiotic use for these two groups (rule-out sepsis and culture negative sepsis) separately. 

The first section of the discussion should briefly summarize your results and what study adds to the field. 

Line 336: did the overall number of sepsis cases really remained unchanged? Was there not an increase in CoNS sepsis? 

The maintenance period was during the COVID-19 pandemic. Which additional actions were implemented at the NICU at that time, that were different from previous periods? Stricter visitation rules – no siblings, only one parent, face masks,… ? Some of them can be considered as AS measures so they should be described more in detail.

Line 347: treatment at birth should be replaced with immediately after birth, as at birth might give the impression you are referring to IAP. 

Lines 353 – 361: there are studies that have showed a reduction in rates of AB exposed preterm infants, some of these and their AS actions should be named here (e.g. Tagare et al. 2010, Bhat et al. 2018, Kitano et al. 2019, Stritzke et al. 2022).

Line 369 – worse should be replaced with higher.

At least 10 out of 30 references are publications published by the same authors. On the other hand, I am missing some important references using a 48-hour stop or other AS actions to reduce the duration of empiric AB in preterm infants (Astorga et al. 2018, Tolia et al. 2017, Lu et al. 2019, Ting et al. 2019, Bhat et al. 2018, Kitano et al. 2019), and other important papers in this field (e.g. Rajar et al. 2020).

The paper has one very clear and important message that implementation of antibiotic stewardship programs targeting premature infants should be considered in all neonatal intensive care units, and results from studies such as this one are important to both encourage and reassure other clinicians in such endeavors. 

Overall, the manuscript is well written, however there are several sentences with repetitive information. There are several sections in the manuscript that lack clarity and sections that should be shortened to be more concise and clearer.

Round 2

Reviewer 3 Report

Comments and Suggestions for Authors

The authors took into account all suggestions and supplemented the paper accordingly. I have no additional comments.